# Preserving inhibition with a disinhibitory microcircuit in the retina

**Qiang Chen[1†], Robert G Smith[2]\*, Xiaolin Huang[3], Wei Wei[1,3,4,5]\***

[1]Committee on Computational Neuroscience, University of Chicago, Chicago, United States; [2]Department of Neuroscience, University of Pennsylvania, Philadelphia, United States; [3]Committee on Neurobiology, University of Chicago, Chicago, United States; [4]Department of Neurobiology, the University of Chicago, Chicago, United States; [5]Grossman Institute for Neuroscience, Quantitative Biology and Human Behavior, University of Chicago, Chicago, United States

**Abstract** Previously, we found that in the mammalian retina, inhibitory inputs onto starburst amacrine cells (SACs) are required for robust direction selectivity of On-Off direction-selective ganglion cells (On-Off DSGCs) against noisy backgrounds (Chen et al., 2016). However, the source of the inhibitory inputs to SACs and how this inhibition confers noise resilience of DSGCs are unknown. Here, we show that when visual noise is present in the background, the motion-evoked inhibition to an On-Off DSGC is preserved by a disinhibitory motif consisting of a serially connected network of neighboring SACs presynaptic to the DSGC. This preservation of inhibition by a disinhibitory motif arises from the interaction between visually evoked network dynamics and short-term synaptic plasticity at the SAC-DSGC synapse. Although the disinhibitory microcircuit is well studied for its disinhibitory function in brain circuits, our results highlight the algorithmic flexibility of this motif beyond disinhibition due to the mutual influence between network and synaptic plasticity mechanisms.

**\*For correspondence:**
rob@retina.anatomy.upenn.edu (RGS);
weiw@uchicago.edu (WW)

**Present address:** †Department of Physiology & Biophysics, University of Washington, Seattle, United States

**Competing interests:** The authors declare that no competing interests exist.

## Introduction

Neural circuits exhibit remarkable complexity and specificity of their wiring patterns. A major effort in neuroscience is to search for unifying principles of neural computation that can be used to analyze and predict input-output relationships of diverse brain circuits. One approach to achieve this goal is to dissect complex circuitry into elementary building blocks, often termed microcircuit motifs (*Braganza and Beck, 2018*; *Cajal, 1937*). These motifs consist of a small number of neurons that are connected in characteristic patterns and are thought to perform defined algorithmic functions in neuronal signal processing across brain regions. With this paradigm, a given large-scale circuit can be analyzed by mapping the computations of its individual motifs from which the entire circuit is assembled.

When two inhibitory interneurons and a principal excitatory neuron are serially connected, they form a microcircuit that is commonly designated as a 'disinhibitory' motif (*Figure 1a*). Activation of the first interneuron is thought to suppress the activity of the second interneuron, causing diminished inhibition of the principal neuron (i.e. disinhibition). Disinhibitory microcircuits are prominently involved in sensory processing, learning, and memory in the neocortex and the hippocampus (*Jiang et al., 2013*; *Lee et al., 2013*; *Letzkus et al., 2011*; *Pfeffer et al., 2013*; *Pi et al., 2013*), and in action selection in basal ganglia (reviewed in *Chevalier and Deniau, 1990*; *Letzkus et al., 2015*). Their functions in these diverse circuits have so far been exclusively attributed to their disinhibitory influences on the principal neurons by relieving the principal neurons from ongoing inhibition.

In the direction-selective circuit of the mammalian retina, serially connected inhibitory interneurons presynaptic to the direction-selective ganglion cell (DSGC) have been identified functionally and anatomically. On-Off type DSGCs, a major type of retinal output neuron with directionally tuned spiking activity (*Barlow and Levick, 1965*), receive directional inhibition from starburst amacrine cells (SACs) (*Briggman et al., 2011*; *Euler et al., 2002*; *Fried et al., 2005*; *Lee et al., 2010*; *Taylor and Vaney, 2002*; *Wei et al., 2011*). SACs themselves receive inhibitory inputs from neighboring SACs (*Chen et al., 2016*; *Ding et al., 2016*; *Kostadinov and Sanes, 2015*; *Lee and Zhou, 2006*) and wide-field amacrine cells (WACs) (*Chen et al., 2016*; *Ding et al., 2016*; *Huang et al., 2019*), and therefore participate in WAC-SAC-DSGC and SAC-SAC-DSGC 'disinhibitory' motifs (*Figure 1b*). When both these motifs are disrupted in a conditional knockout mouse line in which GABA-A receptor a2 subunit is selectively removed from SACs (*Gabra2* cKO) (*Auferkorte et al., 2012*; *Figure 1d* schematic), the direction selectivity of On-Off DSGCs is differentially affected in a visual stimulus-dependent manner (*Chen et al., 2016*). The responses of On-Off DSGCs to the leading edge of a bright moving bar (On responses) in *Gabra2* cKO mice show normal direction selectivity when the bar moves against a homogeneous gray background. However, when the bar moves against a 'noisy' background of a randomly flickering checkerboard, DSGC On responses show impaired direction selectivity (*Chen et al., 2016*). Therefore, in the On pathway, inhibitory input to SACs is not required for the implementation of direction selectivity, but is specifically required for the robustness of this selectivity when flickering noise is present in the motion background. However, the microcircuit motif, as well as the underlying neural mechanism of this noise resilience, is not known.

In this study, we identified the SAC-SAC-DSGC microcircuit as the motif that underlies the noise resilience of direction selectivity in the On pathway. Combining genetic, functional and computational modeling approaches, we found that instead of disinhibiting DSGCs, this motif preserves motion-evoked DSGC inhibition in the noisy background, thereby preserving the direction selectivity of the DSGC. This unexpected, inverted algorithm leverages the center-surround receptive field (RF) of SACs, and results from the interaction between visual noise-generated network dynamics and short-term plasticity at the SAC-DSGC synapse. Our study therefore highlights the flexibility of neural computations by well-defined circuit components, which can be dramatically influenced by interactions of network activity patterns and synaptic plasticity mechanisms.

## Results

### Effect of removing a disinhibitory motif

In *Gabra2* cKO mice, GABAergic inputs to SACs are selectively eliminated while the rest of retinal inhibitory circuitry remains intact (*Chen et al., 2016*; *Figure 1d* schematic). When probed with a moving bar against a gray background, the direction selectivity of On-Off DSGC On spiking responses in *Gabra2* cKO mice was not impaired compared to that in control mice (*Chen et al., 2016*; *Figure 1c–e*). However, when the background of the moving bar was a randomly flickering checkerboard (see Materials and methods), motion-evoked On spiking activity was increased in the anti-preferred, or null direction, resulting in reduced direction selectivity as measured by the direction selectivity index (DSI) (*Figure 1c–d and f*, *Figure 1—figure supplement 1*). This reduced direction selectivity in *Gabra2* cKO mice was observed over a range of checkerboard intensities but was not observed in homogeneous featureless backgrounds of similar intensities or bar contrasts (*Chen et al., 2016*). The enhanced null-direction spiking triggered by the moving bar in the DSGCs of the *Gabra2* cKO group was due to attenuated inhibitory postsynaptic currents (IPSCs) of DSGCs (*Figure 1g–j*), which come from the GABAergic outputs of SACs (*Briggman et al., 2011*; *Fried et al., 2002*; *Lee et al., 2010*; *Taylor and Vaney, 2002*; *Wei et al., 2011*). But in the baseline period before the onset of the moving bar, the flickering checkerboard-evoked DSGC IPSCs in the control and cKO groups had similar amplitudes (*Figure 1—figure supplement 1*). In addition, the excitatory postsynaptic currents (EPSCs) of the DSGC, which originate predominantly from glutamatergic bipolar cells during this noisy bar stimulus (*Figure 1—figure supplement 2*), were not affected compared to the control group (*Figure 1—figure supplement 3*, also see Supplemental

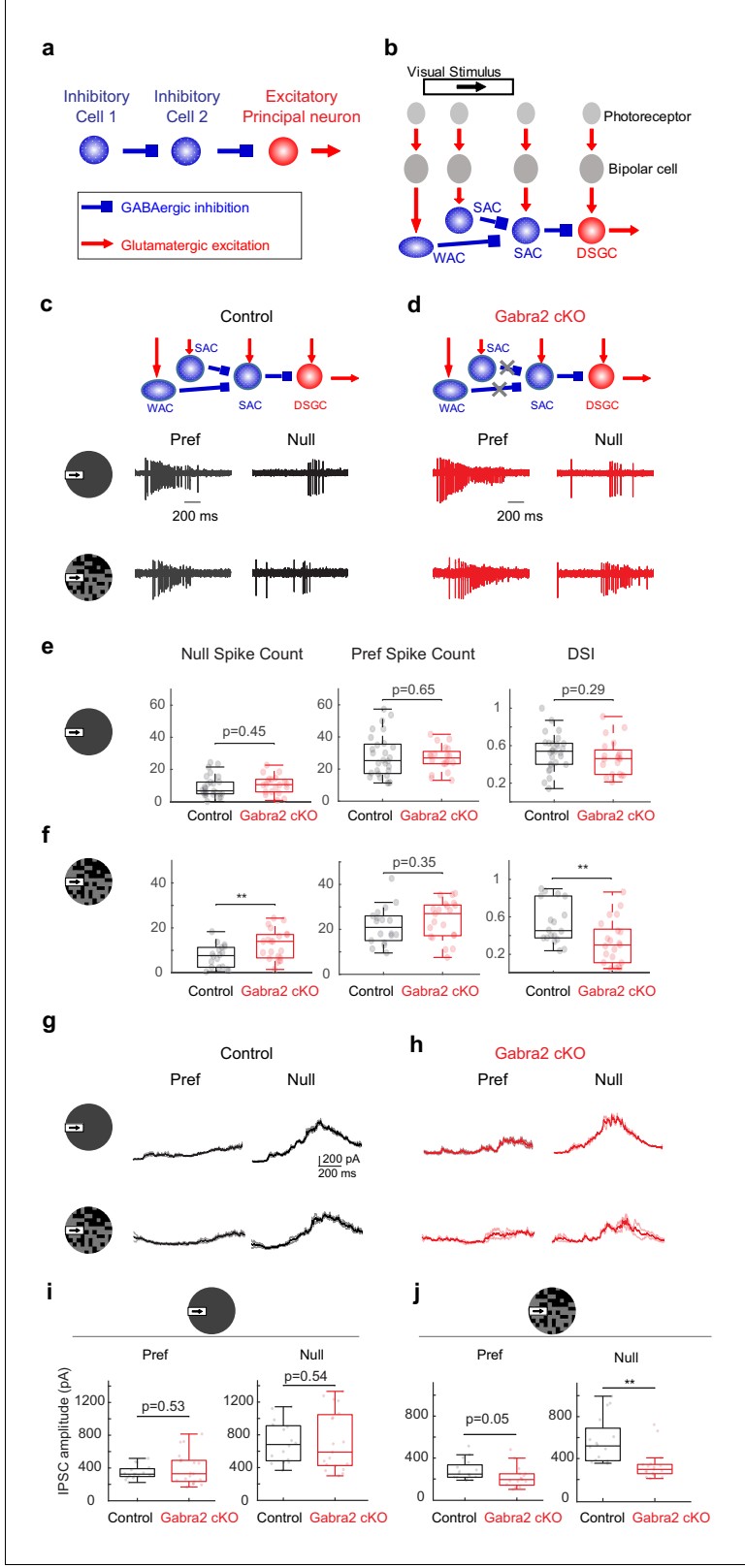

**Figure 1.** On-Off DSGCs in *Gabra2* cKO mice show reduced direction selectivity and reduced null-direction inhibition in the noisy background. (a) Organization of a canonical disinhibitory microcircuit. (b) Schematic of disinhibitory motifs WAC-SAC-DSGC and SAC-SAC-DSGC in the direction-selective circuit. WAC, SAC, and DSGC receive light-evoked glutamatergic inputs from the photoreceptor-bipolar cell signaling pathway. DSGC axons

*Figure 1 continued*

leave the retina and innervate higher brain nuclei. (**c-f**) DSGC spiking activity (**c**) Upper: schematic of disinhibitory motifs in control mice. Lower: example spiking activity of a DSGC in the control group during a moving bar in the preferred (pref) and null directions in noise-free (upper traces) and noisy (lower traces) backgrounds. (**d**) Same as c, from a DSGC in the Gabra2 cKO group. (**e**) Summary plots of null-direction (left), preferred-direction (middle) spike counts and direction selectivity index (DSI, see Materials and methods) in control and Gabra2 cKO groups for moving bar in the noise-free background. Box plots represent minimum / first quartile / median / third quartile / maximum values for this and subsequent figures, dots represent individual cells. Control: n = 19 cells from 6 mice. Gabra2 cKO: n = 20 cells from 6 mice. (**f**) Same as (**e**) for moving bar in the noisy background. (**g-j**) DSGC IPSC . (**g**) Example bar-evoked IPSC traces of a DSGC in the control group during a moving bar in the preferred and null directions in noise-free (upper traces) and noisy (lower traces) backgrounds. Three individual trials (thin lines) and the mean (thick line) are shown. (**h**) Same as g, from a DSGC in Gabra2 cKO group. (**i**) Summary plots of preferred- (left) and null-direction (right) IPSC peak amplitudes in control and Gabra2 cKO groups for moving bar in the noise-free background. Control: n = 17 cells from 6 mice. Gabra2 cKO: n = 20 cells from 7 mice. (**j**) Same as I, for moving bar in the noisy background. See also *Figure 1—figure supplements 1–3*.

The online version of this article includes the following source data and figure supplement(s) for figure 1:

**Source data 1.** On-Off DSGCs in *Gabra2* cKO mice show reduced direction selectivity and reduced null-direction inhibition in the noisy background.
**Figure supplement 1.** Flickering checkerboard-evoked DSGC response.
**Figure supplement 1—source data 1.** Flickering checkerboard-evoked DSGC response.
**Figure supplement 2.** Excitatory inputs onto DSGCs in noisy background are primarily glutamatergic.
**Figure supplement 2—source data 1.** Excitatory inputs onto DSGCs in noisy background are primarily glutamatergic.
**Figure supplement 3.** On-Off DSGC EPSCs are not affected in *Gabra2* KO during the moving bar stimulus in the noisy background.
**Figure supplement 3—source data 1.** On-Off DSGC EPSCs are not affected in *Gabra2* KO during the moving bar stimulus in the noisy background.

Discussion on the contributions of glutamatergic and cholinergic excitation to DSGC with noise-free and noisy backgrounds). Therefore, GABAergic inputs onto SACs are important for the noise resilience of DSGC direction selectivity by preserving the strength of directionally tuned inhibition from SACs to DSGCs during motion over a noisy background.

## Surround suppression of SAC by neighboring SACs

Since GABAergic inhibition of SACs represents the canonical 'disinhibitory' motif for the principal neuron DSGC, one would predict that inhibition of the DSGC by SACs would be enhanced when GABAergic inputs to SACs are removed in *Gabra2* cKO mice. But contrary to this prediction, we have found that removing SAC inhibition in *Gabra2* cKOs leads to diminished null-direction inhibition of DSGCs in the noisy background (*Figure 1j*). To further investigate this unexpected effect, we recorded the somatic membrane potential ($V_m$) of SACs during moving bar stimuli with noise-free and noisy backgrounds. In control animals, the bar entering the SAC's RF surround with a noise-free background evoked a brief period of membrane hyperpolarization prior to the depolarizing response evoked by the bar crossing the RF center (*Figure 2a*, left trace). Similar patterns of SAC surround suppression have been observed in the rabbit (*Lee and Zhou, 2006*). In the presence of flickering noise, control SACs exhibited flicker-evoked membrane depolarization (termed 'flicker response') during the baseline period before the onset of motion (*Figure 2a*, right trace). This flicker response was transiently suppressed when the bar traveled across the RF surround, creating a 'silent', or noise-free period of SAC $V_m$ immediately before motion-evoked depolarization caused by the bar moving in the RF center (*Figure 2a*, right trace).

In *Gabra2* cKO mice, SAC resting somatic membrane potentials and flicker responses during the baseline period were not affected compared to the controls (*Figure 2b, d and g*). The SAC depolarization evoked by the bar traversing the RF center was also not affected in *Gabra2* cKO mice (*Figure 2f and i*). However, moving bar-evoked surround suppression of SAC $V_m$ was significantly impaired. In the noise-free background, SACs in the *Gabra2* cKO group exhibited reduced membrane hyperpolarization (*Figure 2e*). In the noisy background, flicker responses of SACs were less effectively suppressed in the cKO group when the bar traversed the RF surround (*Figure 2h*).

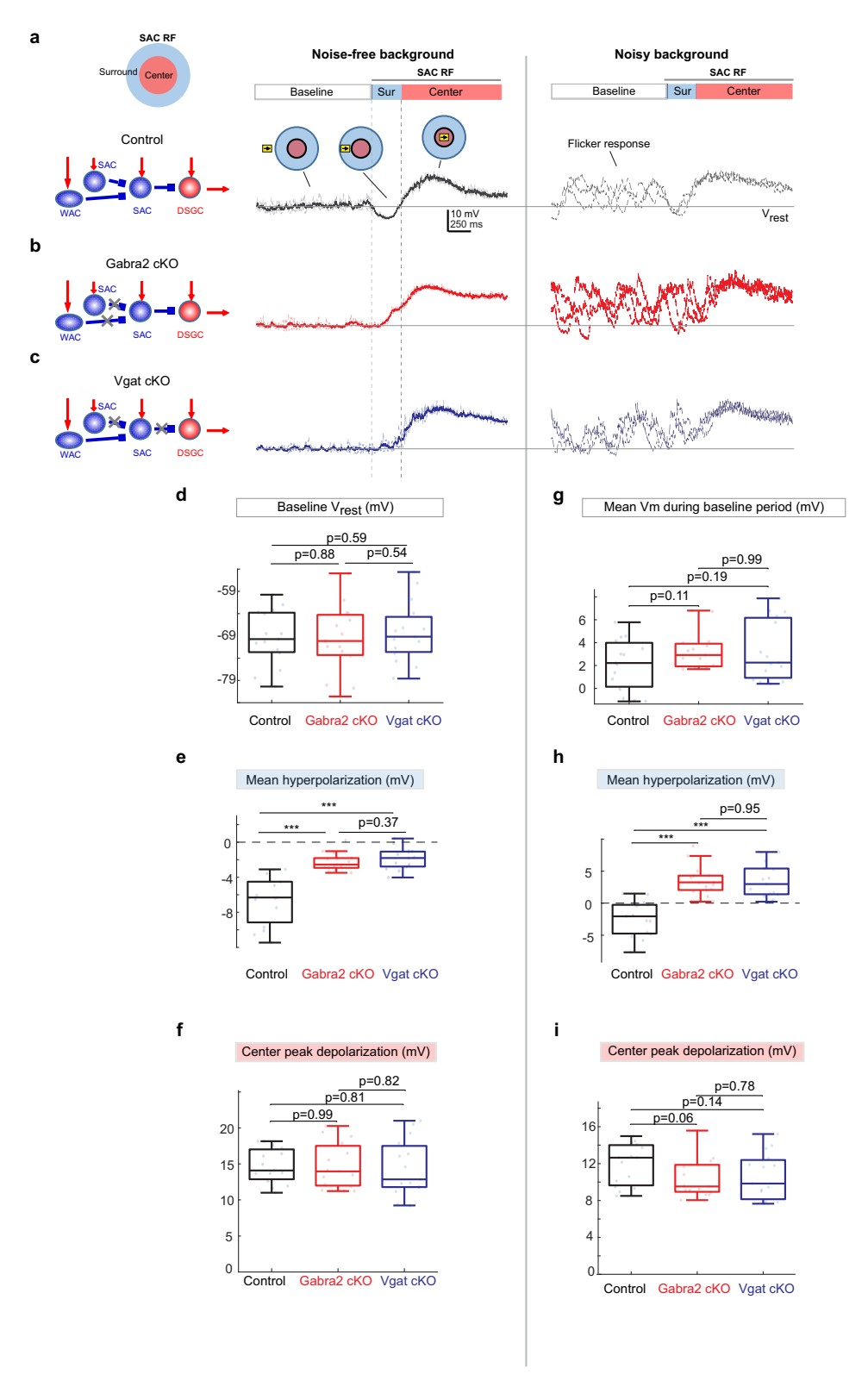

**Figure 2.** Weakened motion-evoked surround suppression of SACs in *Gabra2* cKO and Vgat cKO mice. (a) Example Vm traces from a control SAC during moving bar in noise-free (left) and noisy background (right). For noise-free condition, lighter traces are three individual sweeps and darker trace represents the mean. For noisy condition, three individual sweeps are shown. For each sweep, the flickering checkerboard pattern was randomly generated and differed from those of other sweeps. Horizontal line indicates the resting membrane potential $V_{rest}$ calculated as the mean baseline Vm

*Figure 2 continued on next page*

*Figure 2 continued*

under the noise-free condition. Vertical dashed lines mark the surround suppression time window. Insets: Upper left: schematic of the center-surround RF structure of a SAC. Upper right: the time windows during which the bar moved outside the RF ('Baseline'), in the RF surround (Sur) and center ('Center'). (b) Same as a, from a SAC in the *Gabra2* cKO group. (c) Same as a, from a SAC in the Vgat cKO group. (d-f): summary plots for noisy-free background.d.Mean Vm during the Baseline time window indicated in a. In noisy-free background it is equivalent to the resting membrane potential Vrest.e.Mean hyperpolarization calculated as the mean Vm during the Surround time window relative to Vrest.f.Peak depolarization calculated as the peak Vm during the Center time window relative to Vrest. (g-i): summary plots for noisy backgroundg.Mean Vm during the Baseline window relative to Vrest.h.Mean Vm during the Surround window relative to Vrest.i.Peak Vm during the Center time window relative to Vrest.Control, n = 15 cells from four mice. Gabra2 cKO, n = 15 cells from four mice. Vgat cKO, n = 15 cells from four mice. See also *Figure 2—figure supplement 1*.

The online version of this article includes the following source data and figure supplement(s) for figure 2:

**Source data 1.** Weakened motion-evoked surround suppression of SACs in*Gabra2*cKO and Vgat cKO mice.
**Figure supplement 1.** Weakened motion-evoked surround suppression of SACs in *Gabra2* cKO and Vgat cKO mice during contracting ring stimulus.
**Figure supplement 1—source data 1.** Flickering checkerboard-evoked DSGC response.

Therefore, GABAergic inputs onto SACs play an important role in generating motion-evoked surround suppression of SACs that transiently prevents visual noise-evoked SAC activation prior to their motion-evoked activation.

We next examined the source of inhibitory inputs to the SAC that suppresses SAC flicker responses. To determine if the strong surround suppression of SACs by the moving bar stimulus is caused by inhibition from neighboring SACs or from WACs, we compared SAC somatic $V_m$ waveforms from *Gabra2* cKO mice with those from another cKO line in which the vesicular GABA transporter is selectively deleted in SACs (Vgat cKO, see Materials and methods for genetic details) (*Pei et al., 2015*). Unlike the *Gabra2* cKO in which all GABAergic inputs to SACs are removed, only the SAC-SAC inhibition, but not the WAC-SAC inhibition, is impaired in the Vgat cKO mouse line (*Figure 2c* schematic). We found that compared to the controls, SACs in Vgat cKO mice exhibited weaker surround suppression and enhanced flicker response before the motion response, a deficit similar to that observed in *Gabra2* cKO mice (*Figure 2b–i*). Weaker surround suppression was also observed in the Vgat cKO and *Gabra2* cKO during a contracting ring stimulus (*Figure 2—figure supplement 1*). Therefore, we attribute the strong SAC surround suppression evoked by the moving bar mainly to inhibitory inputs from neighboring SACs.

## SAC RF surround prevents synaptic depression

Why does the blockade of inhibitory inputs to SACs attenuate motion-evoked inhibition of the DSGC with a noisy background? We formulated a working hypothesis based on the following two observations described in the previous section: (1) SACs in *Gabra2* cKO mice show increased flicker-evoked activation during the time window immediately before their motion-evoked responses and (2) motion-evoked SAC depolarization is not impaired in the *Gabra2* cKO mice. Our hypothesis is that the persistent flicker response of the SAC before its motion-evoked depolarization in the *Gabra2* cKO mouse induces short-term synaptic depression at the GABAergic SAC-DSGC synapses, and thereby attenuates moving bar-evoked inhibition of the DSGC. Short-term depression at the SAC-DSGC GABAergic synapse has been demonstrated in P7-14 animals during development using paired pulse stimulation (*Morrie and Feller, 2015*). To examine if this synaptic depression also occurs in the mature retina, we performed paired voltage clamp recordings from SAC-DSGC pairs in adult control mice. SACs were depolarized with a short voltage step, which led to the opening of voltage-gated calcium channels (VGCCs) shown as an inward peak that was time-locked with the postsynaptic response of DSGCs and was blocked by VGCC antagonists (*Lee et al., 2010*; *Koren et al., 2017*). The IPSCs of DSGCs showed pronounced short-term depression in the paired pulse protocol (*Figure 3*), indicating that the SAC-DSGC synapse is prone to depression by repeated SAC activation.

Next we asked if flicker-evoked SAC membrane depolarization in *Gabra2* cKO that persisted into the surround suppression time window can trigger short-term synaptic depression and reduce the subsequent moving bar-evoked GABA release from SACs. To address this question, we performed paired SAC-DSGC somatic recordings in control mice during which the presynaptic SAC was voltage clamped according to waveforms that mimic the activation patterns of control and cKO SACs during moving bar stimuli. Since the neurotransmitter release sites of the axonless SAC are located in the

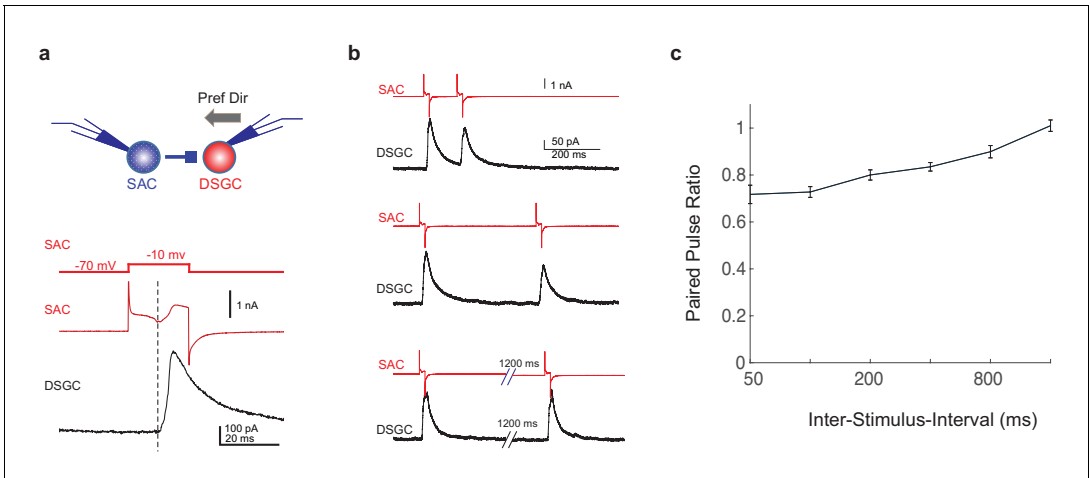

**Figure 3.** GABAergic SAC-DSGC synapses in adult mice exhibit paired pulse depression. (a) Upper: schematic of paired voltage clamp recordings between a DSGC and a SAC on the null side of the DSGC. Preferred direction of the DSGC is shown as the black arrow. Lower: command potential of the SAC (upper trace), holding current of the SAC (middle trace), and DSGC IPSC evoked by SAC depolarization (lower trace). Dashed line indicates the onset of the IPSC, which coincides with the inward calcium current activated in the SAC (*Koren et al., 2017*). (b) Example traces of SAC and DSGC recordings at different inter-pulse intervals (200, 400, 1600 ms). (c) Summary plot of paired pulse ratio (2nd/1st peak amplitudes) as a function of inter-pulse interval. X-axis is in log scale. N = 17 pairs from three mice.

The online version of this article includes the following source data for figure 3:

**Source data 1.** GABAergic SAC-DSGC synapses in adult mice exhibit paired pulse depression.

distal dendrites, we first used computational modeling to estimate $V_m$ at the SAC distal dendrites based on SAC somatic $V_m$ recordings during visual stimulation. A biophysical model of the SAC was constructed based on the digitized morphology of a typical mouse SAC, bipolar cell inputs mapped by connectomic analysis (*Ding et al., 2016*) and experimentally measured SAC membrane properties (*Stincic et al., 2016*; *Figure 4a*, see Materials and methods). Simulation of SAC activation during a moving bar over the flickering checkerboard indicated that the peak amplitudes of the somatic and distal dendritic $V_m$ are comparable (*Figure 4b*). Next, we modeled the degree of attenuation between somatic and distal dendritic compartments during our voltage clamp experiments using parameters that approximate the voltage clamp conditions in the absence of sodium and potassium conductances. Due to the electrotonic isolation of distal SAC dendrites from the soma (*Euler et al., 2002*; *Koren et al., 2017*; *Tukker et al., 2004*), we estimated the membrane depolarization at SAC dendritic tips by computational modeling to be about 25% of somatic depolarization in the whole-cell voltage-clamp configuration used in our study (*Figure 4c*). Therefore, we scaled the somatic $V_m$ traces obtained during current clamp recordings by a factor of 4. The scaled waveforms were then used as the command potentials of SACs in paired voltage-clamp recording experiments so that the $V_m$ at distal SAC dendrites could be depolarized to a level comparable to that during visual stimulation. The IPSCs of the DSGC in each SAC-DSGC pair were recorded to measure SAC GABA release.

For each SAC-DSGC pair, the SAC was depolarized according to two waveforms in random sequences. One was a scaled waveform of a representative somatic $V_m$ recording from a *Gabra2* cKO SAC during the moving bar stimulus over a randomly flickering checkerboard (designated 'cKO pattern', *Figure 5a*, upper red trace). The other waveform was identical to the above cKO pattern except during the time window preceding the motion-evoked response (*Figure 5a*, time window 2) to include the surround suppression period. During this time window, the noisy $V_m$ pattern of the cKO SAC was replaced by the mean $V_m$ pattern of SACs in the control group. The second waveform is termed 'control pattern' (*Figure 5a*, upper black trace). For each SAC-DSGC pair, the cKO and the control $V_m$ waveforms were randomly chosen to voltage-clamp the SAC somatic $V_m$, while the DSGC IPSCs were recorded simultaneously (*Figure 5a*). Since the flicker-induced membrane fluctuations during the baseline period are identical for cKO and control patterns, the DSGC IPSCs during this baseline period (*Figure 5a*, time window 1) served as an internal control for each pair for

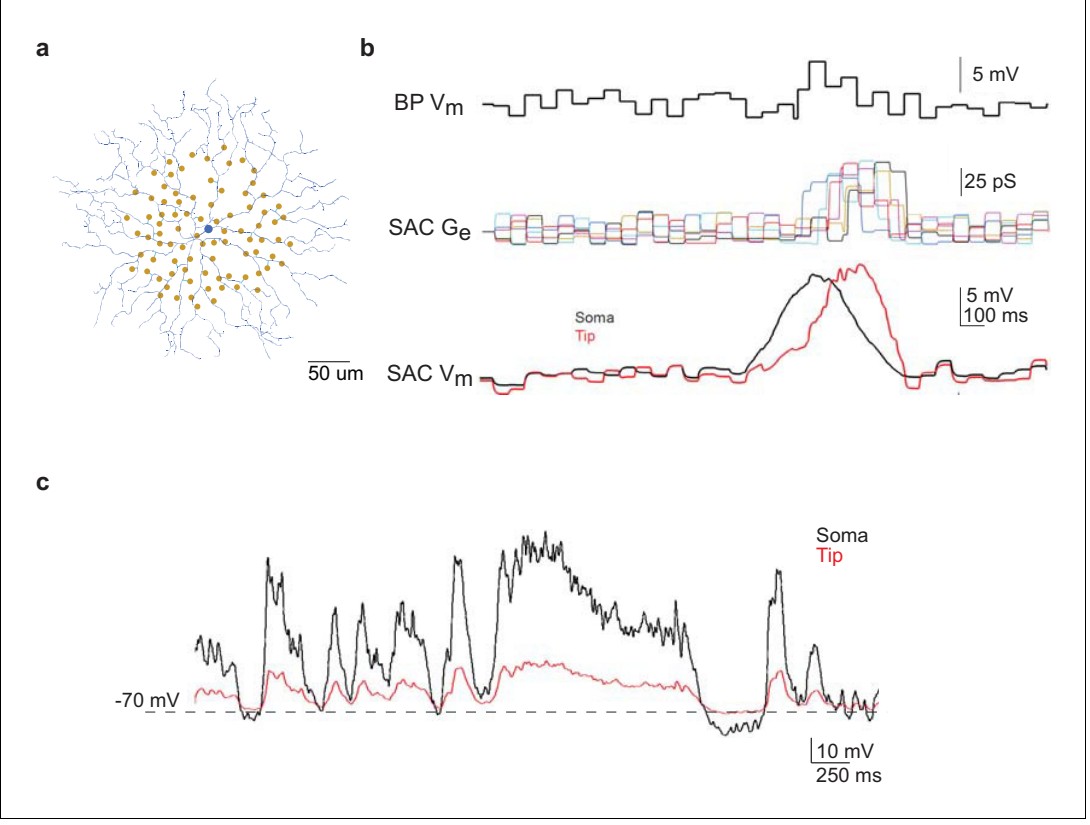

**Figure 4.** Computational modeling of SAC light responses at soma and dendritic tips during the moving bar stimulus in the noisy background. (**a**) A digitized SAC (blue) and locations of bipolar cell somas (yellow dots) used for simulation. (**b**) Upper trace: simulated Vm of a bipolar cell during moving bar stimulus in noisy background. Middle traces: simulated excitatory conductance (Ge) of the SAC evoked by six example bipolar cell inputs. Colored traces are SAC Ge evoked by individual bipolar cells. Lower traces: simulated SAC somatic Vm at the soma and at the distal tip of a dendrite. (**c**) Simulation of soma-to-dendritic tip attenuation in a SAC during voltage clamp recording. Black trace: scaled somatic Vm waveform (black trace) of a SAC based on the current clamp recording of a SAC Vm during the moving bar stimulus in noisy background. Red trace: simulated Vm at the SAC's distal dendritic tip in the voltage-clamp configuration.

ensuring stable recordings and monitoring potential run-down of whole-cell recordings. As expected, the DSGC IPSC amplitudes during the baseline period for the cKO and control patterns were similar (*Figure 5a and b*, time window 1). However, DSGC IPSCs triggered by the motion-evoked SAC waveforms (*Figure 5a and b*, time window 3) were significantly different between the cKO and the control patterns. Even though the SAC was activated by the same motion-evoked depolarization pattern during time window 3, a more active Vm immediately before motion-evoked depolarization (*Figure 5a and b*, time window 2) in the cKO waveform led to smaller DSGC IPSC peak amplitudes evoked by the motion waveform (*Figure 5a–d*, time window 3). We compared the IPSCs in this paired recording protocol with those during the noisy bar visual stimulation. The peak amplitude and charge transfer of IPSCs in paired recordings are in the lower end of the range for DSGC IPSCs evoked by the noisy bar stimulus (charge transfer: light-evoked: preferred direction: 95.1+ / - 8.6 pA*s; null direction, 198.6 + / - 22.5 pA*s, 16 cells; paired recordings: 100+ 12.1 pA*s, 22 cells; amplitude: light-evoked: preferred direction: 284.2 + / - 23.2 pA; null direction, 570.3 + / - 52.7 pA; paired recordings: 174.5 + / - 17.9 pA), supporting that the measured DSGC inhibitory currents in the paired recordings are within the physiological range. Furthermore, the synaptic suppression observed in the cKO group was robust when we tested additional scale factors of 3 and 5 for SAC command waveforms (*Figure 5d*).

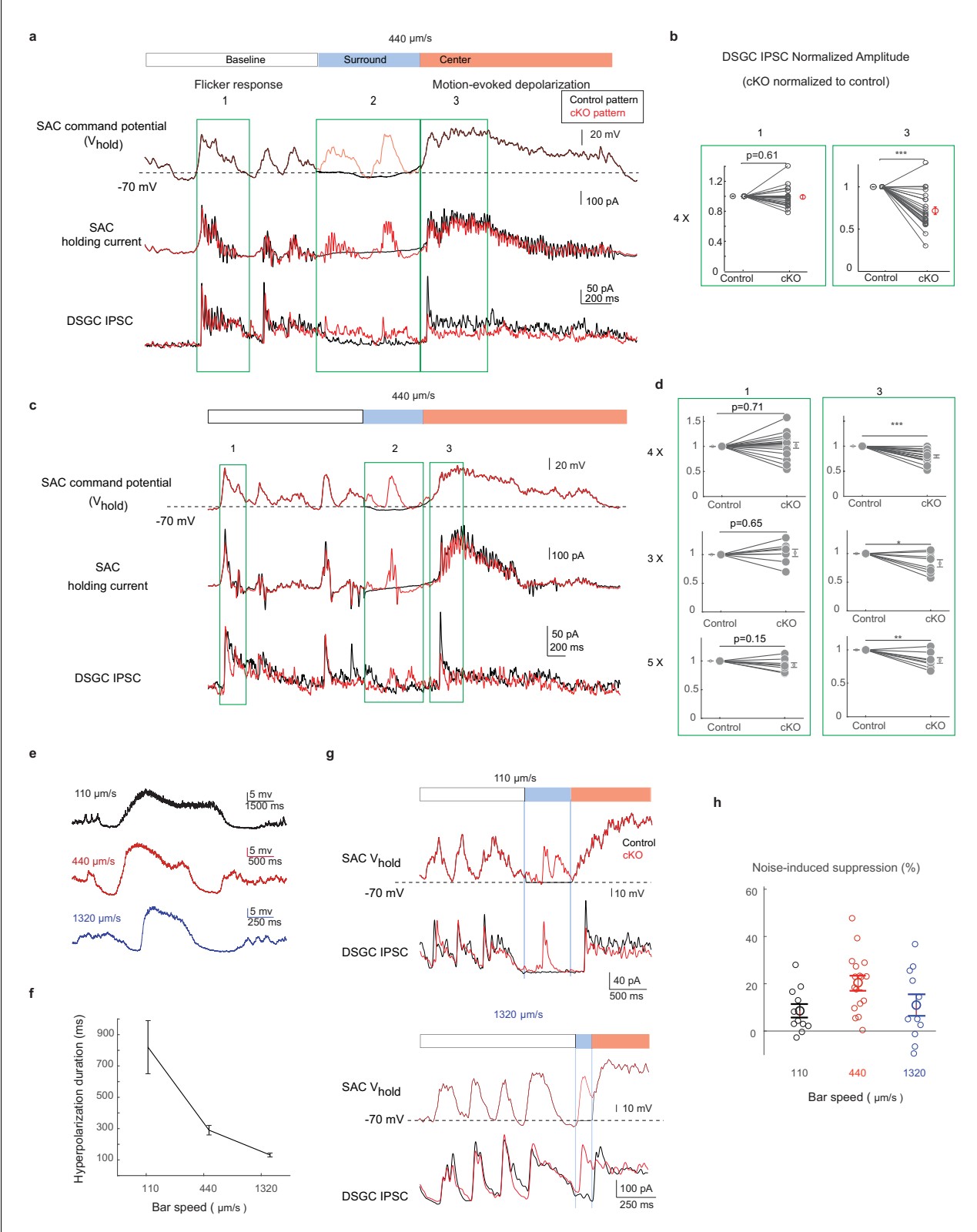

**Figure 5.** The activation pattern of *Gabra2* cKO SACs during motion in the noisy background induces synaptic depression between SACs and DSGCs in control mice. (a) Top traces: the scaled control (black) and *Gabra2* cKO (red) waveforms during noisy bar stimulus used to voltage-clamp SAC somatic Vm in control mice. Middle traces: holding currents of a SAC during the paired SAC-DSGC voltage-clamp recordings. Downward deflections represent SAC calcium currents activated by membrane depolarization. Lower traces: DSGC IPSCs evoked by the control and the cKO waveforms.
*Figure 5 continued on next page*

*Figure 5 continued*

Schematic on top of the traces indicates time windows corresponding to the baseline period, bar-evoked surround suppression and bar-evoked depolarization during current clamp experiments in *Figure 2*. Green boxes numbered 1, 2, and 3 are regions in baseline, surround and center time windows chosen for subsequent quantification. (b) Pair-wise comparisons of DSGC IPSC amplitude evoked by the scaled (4x) control and cKO waveforms in time windows 1 and 3. For each cell pair, the IPSC peak amplitude evoked by the cKO waveform is normalized to that of the control waveform for each time window. N = 17 pairs from seven mice. (c) Same as a, with a different representative cKO waveform. (d) As b, but with different scale factors for the SAC command waveforms in c. (e) Example control SAC Vm traces during moving bar stimuli over gray background at various speeds. (f) Duration of the hyperpolarization window during noise-free moving bar as a function of bar speed. 110 µm/s: 11 cells, 4 mice, 440 µm/s: 15 cells, 4 mice, 1320 µm/s: 21 cells, 6 mice. Data are represented as mean ± SEM. (g) Same as a, but with scaled cKO SAC Vm waveforms at 110 µm/s (upper) and 1320 µm/s (lower). For control waveforms, mean membrane potential of the control waveforms replaces the cKO traces in the surround suppression time window. (h) Summary of synaptic suppression of the SAC-DSGC synapse induced by the *Gabra2* cKO waveform relative to the response evoked by the control waveform for various motion speeds. Noise-induced expression is calculated as the percentage decrease of motion-evoked cKO response relative to the control. 110 µm/s: 11 pairs from three mice; 440 µm/s: 17 pairs from seven mice; 1320 µm/s: 11 pairs from three mice. Data are represented as mean ± SEM. See also *Figure 5—figure supplement 1*.

The online version of this article includes the following source data and figure supplement(s) for figure 5:

**Source data 1.** The activation pattern of *Gabra2* cKO SACs during motion in the noisy background induces synaptic depression between SACs and DSGCs in control mice.

**Figure supplement 1.** *Gabra2* cKO SACs show enhanced noise responses prior to the motion-evoked responses at different speeds.

**Figure supplement 1—source data 1.** *Gabra2* cKO SACs show enhanced noise responses prior to the motion-evoked responses at different speeds.

Next, we recorded the SAC Vm during the moving bar stimuli at different speeds and examined the duration of surround suppression as a function of speed. We tested three speeds: 4° (110 µm)/s – at the low end of the mouse DSGC speed tuning curve (*Ding et al., 2016*; *Hoggarth et al., 2015*), 15° (440 µm)/s – near the optimal speed of the DSGC speed tuning curve, and 44° (1320 µm)/s – at the high end of the tuning curve. As expected, we observed speed-dependent duration of SAC surround suppression in the sub-millisecond range over these speeds (*Figure 5e and f*). Under noisy conditions, cKO SACs exhibited enhanced noise response during the surround suppression window at all speeds (*Figure 5—figure supplement 1*). We then performed paired SAC-DSGC recordings in control mice using representative cKO Vm waveforms from SACs during the noisy bar stimuli at different speeds. We found that for all three speeds, when random flickering checkerboard-evoked SAC activation fell in the SAC surround suppression time window, it could lead to reduced motion-evoked DSGC IPSCs (*Figure 5g and h*). These results are consistent with the paired pulse experiment of SAC-DSGC recordings that indicate short-term depression occurs at a comparable time scale (*Figure 3*). Together, these results strongly support the hypothesis that in *Gabra2* cKO mice, flicker-evoked SAC depolarization immediately preceding motion-evoked depolarization induces short-term depression at the SAC-DSGC GABAergic synapse, and thereby leads to reduced DSGC inhibition evoked by the moving bar.

Considering the extensive network of interconnected SACs (*Ding et al., 2016*; *Lee and Zhou, 2006*), one might ask whether synaptic depression attenuates the hyperpolarization of a SAC by its neighboring SACs in the presence of object motion with background noise. Consistent with this hypothesis, we did observe a weakened hyperpolarization of the SAC with the noisy background compared to the noise-free background (*Figure 2e* versus 2 hr). However, even though GABA release from SACs is depressed with a noisy background, object motion in the visual field still triggers sufficient GABA release from many neighboring SAC varicosities to suppress noise responses in the postsynaptic SAC just ahead of the moving object.

## Discussion

Our study revealed an unexpected algorithm of the SAC-SAC-DSGC disinhibitory motif that mediates the noise resilience of retinal direction selectivity. In contrast to the stereotypical disinhibitory function that has been conventionally assumed for this canonical motif, the SAC-SAC-DSGC motif preserves motion-evoked inhibition of DSGCs by preventing the SAC-DSGC GABAergic synapse from acquiring visual noise-induced short-term synaptic depression. This computation arises from the concerted action of three mechanistic components (*Figure 6*): (1) The lateral inhibition between neighboring SACs creates a SAC RF surround that is strongly engaged by visual motion. (2) A

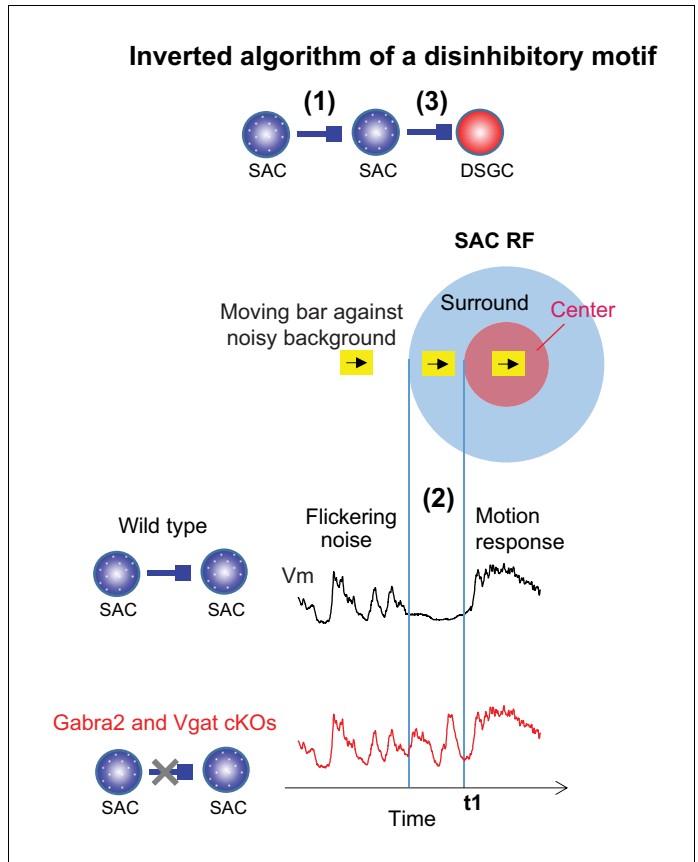

**Figure 6.** Mechanisms underlying the preservation of motion-evoked DSGC inhibition by the SAC-SAC-DSGC disinhibitory motif. The inverted algorithm of the SAC-SAC-DSGC disinhibitory motif arises from three mechanistic components during motion processing in the noisy background. The SAC-SAC inhibition contributes to a strong RF surround of SACs (1), which enables motion-evoked suppression of SAC Vm that dampens SAC flicker response (2). Dampened SAC flicker response before its motion-evoked response prevents short-term synaptic depression of the SAC-DSGC synapse during motion stimuli (3) and preserves the strength of null-direction inhibition of the DSGC (see more in Discussion).

moving object entering the SAC RF surround transiently suppresses visual noise-evoked SAC activation prior to motion-evoked RF center response; (3) The SAC-DSGC GABAergic synapse exhibits short-term synaptic depression on a timescale that matches that of visual noise-triggered network dynamics. Our findings highlight that visual motion stimuli across the center-surround RF elicit characteristic temporal activity profiles in visual neurons that can influence synaptic plasticity mechanisms. The modulation of short-term synaptic plasticity in turn profoundly impacts the computational algorithm of the microcircuit in a context-dependent manner.

What is the operating range of the 'inverted' algorithm of disinhibitory motif in the extensive parameter space of natural motion stimuli? The answer to this question requires a thorough understanding of the interaction between stimulus-driven network dynamics and synaptic plasticity rules of circuit motifs. In this study, noise resilience is explored with a pattern of randomly flickering checkerboard that aims to effectively activate bipolar cells to feed 'noise' into the input end of the motion circuit in the IPL. While many other stimulus conditions await testing, the current set of experiments provides a mechanistic understanding of signal processing by this motif and indicates the SAC surround suppression provides a 'protective' time window to prevent noise-evoked activation from occurring immediately before motion-evoked activation in the SAC, which would otherwise depress its motion-evoked neurotransmitter release. This function does not necessitate that the SAC-DSGC synapse be chronically depressed by the baseline noise before the onset of motion-evoked surround suppression. Therefore, this function does not critically depend on the recovery from existing depression during the surround suppression window.

It is notable that the duration of SAC surround suppression varies with motion speed. At higher speeds, the surround suppression window could be too short to separate the noise response from the motion response, so that synaptic depression could occur even with intact SAC-SAC inhibition in wild-type mice. We reasoned that the more depressed motion-evoked release of *Gabra2* cKO SACs is not only due to less recovery from synaptic depression, but is likely also due to further depression by the extra noise activity within the surround suppression window. When we selected *Gabra2* cKO SAC voltage traces that contained the 'unwanted' noise activation within this shorter window as command waveforms in paired recordings (*Figure 5g*, lower panel), we observed more severe synaptic depression of the SAC-DSGC synapse than with the control waveform. Therefore, the SAC-SAC inhibition is still effective in shielding the SAC from further depression by noise activity within the ~150 ms window prior to the motion response. The persistence of the protective role of SAC-SAC inhibition at the higher speed suggests that the short period immediately before the motion response plays an important role in shaping the motion-evoked SAC transmitter release.

Together, the RF structure and synaptic plasticity of SACs suggest predictable and testable limits of the visual stimulus parameters for the noise resilience function of this disinhibitory motif. While this study does not address the questions of how the SAC-mediated disinhibitory motif impacts visual processing at the population level or the consequences for vision, we believe the results will be useful to aid future studies to address how different visual stimuli trigger different patterns of spatiotemporal network activity at the level of synaptic circuitry, and recruit short-term synaptic plasticity at individual synapses.

In this study, we focused on the On pathway of the direction-selective circuit, which depends on the synaptic interactions between On SACs and the On dendritic layer of the bistratified On-Off DSGCs. In the Off pathway, the SAC-SAC-DSGC motif is also present (*Ding et al., 2016*), and may play a similar role in preserving DSGC direction selectivity under noisy conditions as in the On pathway. However, eliminating inhibitory inputs onto Off SACs in *Gabra2* cKO mice leads to impaired direction selectivity of DSGCs even in the absence of visual noise in the background (*Chen et al., 2016*), making it difficult to unambiguously separate the roles of lateral inhibition motifs in the generation versus the noise resilience of direction selectivity in the Off pathway. Different functional contributions of anatomically symmetric motifs for the direction-selective responses to bright and dark motion stimuli suggest context-dependent neural computations by the 'hardwired' retinal circuitry.

In addition to the inhibition from neighboring SACs, wide-field amacrine cells also contribute to the surround inhibition of SACs (*Ding et al., 2016*; *Lee and Zhou, 2006*). We found that in the absence of SAC-SAC inhibition in Vgat cKO mice, wide-field inhibition of the SAC was not strong enough to significantly suppress the noise response of the SAC in our noisy bar stimulus that was presented to a retinal area of 660 um in diameter. It is possible that more extensive stimuli may trigger stronger wide-field inhibition to further facilitate noise suppression. It has been shown that the wide-field inhibition of SACs is sensitive to the continuity of moving edges, and thereby confers contextual modulation of pDSGC spiking response to compound gratings that differ between center and surround regions (*Huang et al., 2019*). Therefore, wide-field amacrine cell inputs to SACs might mainly serve a different function of detecting discontinuities while the SAC-SAC inhibition mediates strong surround suppression prior to the approaching motion.

The mechanistic components underlying the inverted role of disinhibition in our study are broadly relevant to brain circuitry within and beyond the retina. For example, surround suppression of inhibitory neurons by lateral inhibition is widely observed in both retinal [*Eggers and Lukasiewicz, 2010*] and cortical circuits (e.g. (*Ayaz et al., 2013*; *Fu et al., 2014*; *Pecka et al., 2014*; *Pi et al., 2013*]). The prevalence of disinhibitory motifs, synaptic plasticity at multiple time scales, and context-dependent network dynamics raises the possibility that this motif can implement a richer set of neural computations beyond disinhibition in other brain regions such as cortex and hippocampus (*Pakan et al., 2016*). For example, vasoactive intestinal polypeptide (VIP) + interneurons, which are known to inhibit other inhibitory neurons in the neocortex and hippocampus, have been shown to play important roles in gating and gain control during sensory and cognitive processing. Although the VIP interneuron-mediated disinhibitory motif differs from the SAC-mediated motif in many specific aspects such as cellular physiology, embedded networks, and input signals, the VIP interneuron-mediated motif is subject to the same influence of ongoing network activity and short-term plasticity, and therefore is likely susceptible to similar algorithmic flexibility as depicted mechanistically in the current study. We further postulate that context-dependent computation is likely a general property of

microcircuit motifs in the brain because synaptic plasticity and network activity patterns are universal and fundamental elements of brain circuits that exert mutual influences onto each other. Future studies that explore the interplay between synaptic plasticity and network dynamics in circuit analysis and modeling will improve our understanding and prediction of the input-output relationships of diverse brain circuits.

# Materials and methods

## Key resources table

| Reagent type (species) or resource | Designation | Source or reference | Identifiers | Additional information |
|---|---|---|---|---|
| Strain, strain background (*M. musculus*) | wild-type (C57BL/6J) mice | The Jackson Laboratory | RRID:IMSR_JAX:000664 | |
| Strain, strain background (*M. musculus*) | Chat-IRES-Cre (129S6-*Chat*$^{tm2(cre)Lowl}$/J) mice | The Jackson Laboratory | RRID:IMSR_JAX:006410 | |
| Strain, strain background (*M. musculus*) | floxed tdTomato (129S6-*Gt(ROSA)26 Sor*$^{tm9(CAG-tdTomato)Hze}$/J) mice | The Jackson Laboratory | RRID:IMSR_JAX:007905 | |
| Strain, strain background (*M. musculus*) | Drd4-GFP (SW-Tg(Drd4-EGFP) W18Gsat/Mmnc) mice | MMRRC | RRID:MMRRC_000231-UNC | |
| Strain, strain background (*M. musculus*) | *Gabra2*$^{flox/flox}$ mice | Gift from Dr. Uwe Rudolph, *Auferkorte et al., 2012* | | |
| Strain, strain background (*M. musculus*) | *Slc32a1*$^{t\ flox/flox}$ (*Slc32a1*$^{tm1Lowl}$/J) mice | The Jackson Laboratory | RRID:IMSR_JAX:012897 | |
| Chemical compound, drug | D-AP5 | Tocris | Cat#0106; CAS: 79055-68-8 | |
| Chemical compound, drug | DNQX disodium salt | Tocris | Cat#2312; CAS: 1312992-24-7 | |
| Chemical compound, drug | Dihydro-β-erythroidine hydrobromide (DhβE) | Tocris | Cat#2349; CAS: 29734-68-7 | |
| Chemical compound, drug | L-AP4 | Tocris | Cat#0103; CAS: 23052-81-5 | |
| Software, algorithm | NeuronC Neural Simulation Language | | Data subfolder nc/models/sbac_noise | manual: http://retina.anatomy.upenn.edu/~rob/neuronc.html source code: ftp://retina.anatomy.upenn.edu/pub/nc.tgz |
| Software, algorithm | PCLAMP 10 | Molecular Devices | | https://www.molecular devices.com/systems/conventional-patch-clamp/pclamp-10-software; RRID:BDSC_14352 |
| Software, algorithm | Prairie View | Bruker Technologies | | https://www.bruker.com/products/fluorescence-microscopes/ultima-multi photon-microscopy/ultima -in-vitro/overview.html |
| Software, algorithm | MATLAB | MathWorks | | http://www.mathworks.com; RRID:SCR_01622 |

*Continued on next page*

*Continued*

| Reagent type (species) or resource | Designation | Source or reference | Identifiers | Additional information |
|---|---|---|---|---|
| Other | Custom MATLAB scripts for visual stimulation and data analysis | This paper | | https://github.com/chrischen2/eLife2020Stimulus.git; *Chen, 2020*; copy archived at swh:1:rev:dd7cc7b01d0fd41d62335ceef0f72a5e921cf374 |

## Animals

The C57BL/6 wild-type or transgenic mice of both sexes were used in this study. The *Gabra2* flox/flox mouse line was a generous gift from Dr. Uwe Rudolph at Harvard Medical School. *Slc32a1* (Vgat) flox/flox mice, SAC-specific *Chat-IRES-Cre* mice, and floxed *tdTomato* mice were originally acquired from the Jackson Laboratory. *Drd4–GFP* mice, which specifically labeled the posterior-direction On-Off DSGC subtypes (*Huberman et al., 2009*), were initially developed by MMRRC (http://www.mmrrc.org/strains/231/0231.html) in the Swiss Webster background, and were subsequently backcrossed to C57BL/6 background. All strains in our laboratory were crossed to C57BL/6 background and crossed with each other to create the lines used in the study. Control mice contained Drd4-GFP, Chat-IRES-Cre and floxed tdTomato transgenes. *Gabra2* cKO mice contained Drd4-GFP, Chat-IRES-Cre, floxed tdTomato and homozygous *Gabra2* flox/flox. Vgat cKO mice contained Drd4-GFP, Chat-IRES-Cre, floxed tdTomato and homozygous *Slc32a1*tflox/flox. Mice from postnatal 21 to 60 days were used in the experiments. All procedures to maintain and use mice were in accordance with the University of Chicago Institutional Animal Care and Use Committee (Protocol number ACUP 72247) and in conformance with the NIH Guide for the Care and Use of Laboratory Animals and the Public Health Service Policy.

## Whole-mount retina preparation

The procedures for isolating the retina from the pigment epithelium have been previously described (*Wei et al., 2010*). In short, mice were dark adapted for at least 45 min, anesthetized with isoflurane, and then decapitated. The retina was then isolated under infrared illumination at room temperature in oxygenated (95% $O_2$/5% $CO_2$) Ames' medium (Sigma-Aldrich, St. Louis, MO). Isolated retinas were then cut into dorsal or ventral pieces and mounted on top of a 1–2 mm$^2$ hole in a small piece of filter paper (Millipore, Billerica, MA) with ganglion-cell-layer up. The orientation of the preferred direction (posterior) of *Drd4*-GFP positive cells was labelled for each piece. Retinas were kept in darkness at room temperature in oxygenated Ames' medium until use (0–8 hr).

## Visual stimulation

A white organic light-emitting display (OLEDXL, eMagin, Bellevue, WA; 800 × 600 pixel resolution, 60 Hz refresh rate) was controlled by an Intel Core Duo computer with Windows seven operating system and presented to the retina at 1.1 μm/pixel resolution. Visual stimuli were generated by MATLAB and Psychophysics Toolbox (*Brainard, 1997*), and focused on the photoreceptor layer through the condenser lens of the microscope. A positive-contrast bar (110 μm wide, 385 μm long) moved along the long axis and was presented in 8 or 12 pseudo-randomly chosen directions over an area of 660 μm in diameter. Unless otherwise noted, the bar traveled at a speed of 440 μm/s on the retina, or about 15° visual angle/s. Three to five trials were recorded for each moving direction. The intensity of the moving bar was ~$6.3 \times 10^4$ isomerizations (R$^*$)/rod/s, in the photopic range. For the noise-free background, the intensity of background was ~1800 R$^*$/rod/s, at the lower end of photopic range. For the noisy background, the noise was generated as a randomly flickering checkerboard. Individual checks were 55 μm x 55 μm in size, with intensity at either background (0) or ~$1 \times 10^4$ R$^*$/rod/s drawn from a binomial distribution. The checkerboard pattern was refreshed at 15 Hz. For each sweep, the flickering checkerboard pattern was randomly generated. Therefore, flickering checker board patterns differed between individual repetitions.

## Two-photon guided recording of fluorescence-positive neurons for light response

The retinas were perfused with oxygenated Ames at 33–34°C during recordings. Drd4-GFP positive DSGCs or tdTomato positive SACs were identified using a two-photon microscope (Bruker Nano Surface Division) and a Ti: sapphire laser (Coherent Chameleon Ultra II) tuned to 920 nm. Then cells were visualized with infrared light (>900 nm) and an IR-sensitive video camera (Watec). The inner limiting membrane was removed with an empty glass electrode to expose the targeted cell. For loose-patch recording of spikes, an electrode of 3–5 MΩ was filled with filtered Ames' medium. For voltage-clamp whole cell recording of On-Off DSGCs, the recording electrode was filled with a cesium-based internal solution containing 110 mM CsMeSO$_4$, 2.8 mM NaCl, 4 mM EGTA, 5 mM TEA-Cl, 4 mM adenosine 5'-triphosphate (magnesium salt), 0.3 mM guanosine 5'-triphosphate (trisodium salt), 20 mM HEPES, 10 mM phosphocreatine (disodium salt), 5 mM N-Ethyllidocaine chloride (QX314), 0.025 mM Alexa 488 (for SACs), and 0.025 mM Alexa 594 (for pDSGCs), pH 7.25. Light evoked IPSCs and EPSCs of DSGCs were isolated by holding cells at 0 mV and −60 mV, respectively. Space clamp is an intrinsic limitation of the voltage clamp method. However, voltage clamp recordings of synaptic currents in On-Off DSGCs have been carefully examined by multiple experimental and modeling studies. While space clamp is still an issue at distal dendrites (*Percival et al., 2019*; *Poleg-Polsky and Diamond, 2011*), light-evoked EPSCs and IPSCs of DSGCs measured by somatic voltage clamp recordings have been taken to well approximate at least the proximal synaptic inputs based on pharmacological and conductance analysis (e.g. (*Cafaro and Rieke, 2010*; *Taylor and Vaney, 2002*). For current-clamp whole cell recording of SACs, the recording electrode was filled with a K$^+$ based internal containing 120 mM KMeSO$_4$, 10 mM KCl, 0.07 mM CaCl$_2$·2H$_2$O, 10 mM HEPES, 0.1 mM EGTA, 2 mM adenosine 5'-triphosphate (magnesium salt), 0.4 mM guanosine 5'-triphosphate (trisodium salt), 10 mM phosphocreatine (disodium salt). Liquid junction potentials (~10 mV) for voltage-clamp and ~14 mV for current clamp recordings) were corrected.

Data were acquired using Multiclamp 700B amplifier, and Digidata 1500A digitizer (Molecular Devices, Sunnyvale, CA) and PClamp 10 software. Data were low-pass filtered at 4 kHz, and digitized at a sampling rate of 10 kHz. The On responses of SACs and DSGCs to the leading edge of the bright moving bar were examined in this study. Recording data were analyzed in PClamp and MAT-LAB. The direction selectivity index (DSI) was defined as (pref-null)/(pref+null).

## Modeling

We constructed a model of a mouse SAC and its bipolar cell inputs using the simulation language Neuron-C (*Smith, 1992*). We used a previously digitized mouse SAC morphology and estimated membrane parameters (V$_{rest}$−70 mV, R$_i$ 40 Ωcm, R$_m$ for proximal dendrites 9.2e3 Ωcm$^2$, R$_m$ for distal varicosities 20e3 Ωcm$^2$ and Cm 0.9 μF/cm$^2$) for the SAC model from this and previous studies (*Stincic et al., 2016*). Sodium and potassium channels were blocked (i.e. not included). Bipolar cell inputs were created in a semi-random pattern onto the SAC dendritic tree within 100 μm of the SAC soma based the spacing and subcellular distribution patterns of bipolar cell inputs onto SACs reported in the connectomic analysis (*Ding et al., 2016*). The checkerboard stimulus had a check size of 12.5 μm and the moving bar width along the direction of motion, 110 μm. Bipolar cell activation was simulated as step depolarizations for the duration of the stimuli over the bipolar cell's RF. The membrane potential in the bipolar cells was set 7 mV above the threshold for synaptic release, approximately −45 mV. The bipolar synapses had tonic release and an exponential release function with a gain of 6 mV/e-fold change, and during the noise stimulus they (~10 per SAC dendrite) each had a conductance that varied between 20 and 50 pS. The model computed the SAC dendritic V$_m$ evoked by a bar moving in the centrifugal direction to estimate the SAC dendritic activation levels during motion in the DSGC's null direction. For simulation of V$_m$ attenuation from soma to distal dendrites of SAC during paired voltage-clamp recording, bipolar cell inputs were removed from the model. Models were run on an array of 3.2 GHz AMD Opteron CPUs interconnected by Gigabit ethernet, on the Mosix parallel distributed task system under the Linux operating system. Source code for the Neuron-C simulator is available at: ftp://retina.anatomy.upenn.edu/pub/nc.tgz. The scripts and data for *Figure 4* are in the subfolder nc/models/sbac_noise. The Neuron-C package compiles and runs under Linux and Mac OSX. An Ubuntu Linux image that runs under virtualbox (http://www.virtualbox.org) is available at ftp://retina.anatomy.upenn.edu/pub/ubuntu.vdi.zip.

## Dual whole-cell patch-clamp recording

Dual whole-cell voltage clamp recording between SACs and pDSGCs was performed in oxygenated Ames medium at 33–34˚C in the presence a cocktail of 0.05 mM D-AP5, 0.02 mM DNQX disodium salt, and 0.005 mM L-AP4 to block retinal light responses. Neurons were visualized with transmitted visible light from a halogen bulb. The preferred direction of Drd4-GFP-positive DSGCs was noted for each retina piece. tdTomato-positive SACs and Drd4-GFP-positive pDSGCs were identified with epifluorescence imaging (X-Cite) under a water immersion 60x objective. SACs located on the null side of DSGCs with overlapping dendritic area (inter-soma distance 30–100 μm) were selected for paired recording with DSGCs. Evoked IPSCs onto DSGCs were isolated by holding DSGCs at 0 mV. For paired voltage-clamp recording between SACs and pDSGCs to mimic light-evoked activation of SACs, representative SAC holding potential waveforms that match the average number of noise events and the mean event amplitude were selected from current-clamp recordings of cKO SACs during the noisy bar visual stimulation. The somatic waveforms were scaled to best mimic the membrane depolarization patterns at SACs' distal dendritic tip during visual stimulation (see Results). Only recordings with series resistances < 20 MΩ, and ratio of membrane resistance to series resistance >10 were included. For each SAC-DSGC pair, the peak DSGC IPSC amplitudes evoked by the control SAC Vm waveform in time window one is averaged across trials. This averaged 'control waveform' value is then used as a normalization factor for the averaged peak DSGC IPSC amplitude evoked by the 'cKO' SAC Vm waveform of the same SAC-DSGC pair in time window one across trials. The same normalization method is used for DSGC IPSC peak amplitudes in time window 3. Therefore, for each pair, time window one serves as an internal control for stable recording while time window three reveals the effect of SAC surround suppression on the short-term depression of SAC GABA release.

## Statistical analysis

Grouped data in *Figures 1* and *2* were presented in boxplot, with the central mark indicates the median and the lower and upper edges of the box indicates 25% and 75% of the data, respectively. Grouped data in *Figures 3* and *5* were presented as mean ± S.E.M. Statistical tests were performed for each grouped data using unpaired (*Figures 1* and *2*) or paired (*Figure 5*) student t-tests with post-hoc Bonferroni correction. *$p<0.05$, **$p<0.005$, ***$p<0.0005$.

## Acknowledgements

We thank Chen Zhang for managing mouse colony, Dr. Uwe Rudolph at Harvard Medical School for the generous gift of the *Gabra2*flox/flox mouse line. This work was supported by NIH R01 EY024016, McKnight Scholarship Award to WW, and NIH R01 EY022070 to RGS.

## Additional information

### Funding

| Funder | Grant reference number | Author |
| --- | --- | --- |
| NIH | R01 EY024016 | Wei Wei |
| McKnight Foundation | McKnight Scholarship Award | Wei Wei |
| NIH | R01 EY022070 | Robert G Smith |

The funders had no role in study design, data collection and interpretation, or the decision to submit the work for publication.

### Author contributions

Qiang Chen, Conceptualization, Formal analysis, Investigation, Methodology, Writing - original draft, Writing - review and editing; Robert G Smith, Resources, Software, Formal analysis, Funding acquisition, Investigation, Methodology, Writing - review and editing; Xiaolin Huang, Investigation; Wei

Wei, Conceptualization, Resources, Supervision, Funding acquisition, Writing - original draft, Writing - review and editing

#### Author ORCIDs
Robert G Smith (iD) https://orcid.org/0000-0001-5703-1324
Xiaolin Huang (iD) https://orcid.org/0000-0001-7367-8347
Wei Wei (iD) https://orcid.org/0000-0002-7771-5974

#### Ethics
Animal experimentation: All procedures to maintain and use mice were in accordance with the University of Chicago Institutional Animal Care and Use Committee (Protocol number ACUP 72247) and in conformance with the NIH Guide for the Care and Use of Laboratory Animals and the Public Health Service Policy.

#### Decision letter and Author response
Decision letter https://doi.org/10.7554/eLife.62618.sa1
Author response https://doi.org/10.7554/eLife.62618.sa2

## Additional files
#### Supplementary files
• Transparent reporting form

#### Data availability
All data generated or analysed during this study are included in the manuscript and supporting files. Custom scripts are available at https://github.com/chrischen2/eLife2020Stimulus.git (copy archived at https://archive.softwareheritage.org/swh:1:rev:dd7cc7b01d0fd41d62335ceef0f72a5e921cf374/).

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

# Appendix 1

## Supplementary discussion related to *Figure 1—figure supplements 2 and 3*

Two lines of evidence indicate that bipolar cell-mediated glutamatergic excitation is the major excitatory drive to DSGCs during the moving bar stimulus in the noisy background in our study.

First, based on *Figure 1—figure supplement 2a–2c*, DSGC spiking onset is delayed with noisy background compared to that with noise-free background, which is consistent with diminished contribution of cholinergic inputs from laterally connected SACs. The excitatory inputs to DSGCs come from two sources: bipolar cells and SACs. The cholinergic inputs from the SAC account for the majority of the early phase DSGC excitation because of the large dendritic span (~220 um) of the SAC that forms lateral connections with the DSGC, while the more local bipolar cell-mediated glutamatergic excitation has a delayed onset compared to the cholinergic excitation (*Sethuramanujam et al., 2016*). The contribution of cholinergic inputs to the onset of DSGC spiking has been shown by pharmacological blockade of nicotinic receptors, which results in a delayed onset of DSGC spiking (*Sethuramanujam et al., 2018*).

Second, based on *Figure 1—figure supplement 2d–2e*, in noise-free background, motion-evoked DSGC EPSC amplitude is significantly reduced by the addition of the cholinergic receptor antagonist DHβE, indicating that both cholinergic and glutamatergic inputs contributed to DSGC EPSCs during motion in noise-free background. In contrast, DHβE did not cause significant reduction of EPSCs during motion in the noisy background, indicating that cholinergic contribution was not significant under this stimulus condition. Therefore, in contrast to the noise-free background, when the moving bar is against the flickering checkboard used in our study, bipolar cell-mediated glutamatergic inputs dominate DSGC excitation over SAC-mediated cholinergic inputs. Consistent with the predominant contribution of glutamatergic excitation to DSGCs under the noisy background condition, we did not detect a significant difference of DSGC EPSCs between the control and the *Gabra2* cKO groups (*Figure 1—figure supplement 3*).

