## [Decision Letter]

Thank you for submitting your article "Preserving inhibition with a disinhibitory microcircuit in the retina" for consideration by *eLife*. Your article has been reviewed by three peer reviewers, and the evaluation has been overseen by Andrew King as the Senior Editor and Reviewing Editor. The reviewers have opted to remain anonymous.

The reviewers have discussed the reviews with one another and the Reviewing Editor has drafted this decision to help you prepare a revised submission.

We would like to draw your attention to changes in our revision policy that we have made in response to COVID-19 (https://elifesciences.org/articles/57162). Specifically, when editors judge that a submitted work as a whole belongs in *eLife*, but that some conclusions require a modest amount of additional new data, as they do with your paper, we would normally ask that the manuscript be revised to either limit claims to those supported by data in hand, or to explicitly state that the relevant conclusions require additional supporting data. In this instance, we hope that you will be able to carry out the additional work outlined below in order to bolster the very interesting conclusions of this study.

Summary:

Certain retinal ganglion cells are selective for the direction of motion of visual stimuli, and maintain this selectivity in the presence of a noisy background. Building on previous results, in this paper the authors find the mechanism responsible for this resilience to noise in the stimulus. They show that direction selectivity is preserved thanks to a disinhibitory circuit involving starburst amacrine cells, which leads to recovery from depression at the synapse with direction-selective ganglion cells so that their responses to coherent directional motion are enhanced. This is an insightful contribution, which provides an important mechanistic extension of the previous work and uncovers an interesting mode of operation of the widespread circuit motif of serial inhibition that could be present in other neural systems. It also highlights the challenges associated with maintaining stimulus feature selectivity when dealing with complex stimuli.

Essential revisions:

The reviewers were all enthusiastic about this paper, noting that the results are well supported by the experimental data. They identified certain areas where they felt that there are limitations in the study and made constructive suggestions for how these could be addressed in a timely fashion.

1) The dynamic clamp part is very elegant, but a limitation is that the relation between what happens at the level of the dendritic tip and what is recorded in the soma is complex, especially in starburst amacrine cells (SAC), due to the electrotonic isolation and the differences in selectivity of the different dendrites. When recording from the soma, one looks at an average contribution from the different dendrites, not specifically the ones that connect to the direction-selective ganglion cells of interest. The modelling work of the authors aims at solving this problem, but it is not clear if it really shows that, in such a noisy condition, the voltage in the dendrite will just be the one in the soma up to a scaling factor. Indeed, a major concern among the reviewers is that there is no direct experimental support for the model's prediction that the dendrites attenuate the somatic voltage-clamp signal by a factor of four (Figure 4). This prediction may be essential to interpreting the key experiment. One approach might be a calcium imaging control for voltage attenuation to compare calcium signals in the soma and dendrites in response to somatic depolarization, but that has its own caveats. Another would be simply to apply a range of scaling factors to the command waveform to show that the differences in depression are robust.

2) Figure 3 shows that recovery from synaptic depression takes several hundred milliseconds (at 400 ms, most of the paired-pulse suppression is still there), but faster motion stimuli allow only for 100 to 300 ms hyperpolarization (Figure 5D). Thus, one may expect that the suggested mechanism becomes less effective for higher stimulus speeds and that noise-induced suppression of the SAC output should by observed even with intact SAC-SAC inhibition. A more detailed consideration of the time that it takes for synaptic depression to recover is needed to help support the idea that this is indeed at the root of the observed phenomenon and may help understand the limits of the proposed circuit motif. (As a note aside, Figure 5F shows similar levels of noise-induced suppression for different speeds, but it's not clear to what degree the selected SAC voltage traces are representative enough of the ensemble of noise-induced voltage traces to allow for a quantitative comparison.)

---

## [Author Response]

Essential revisions:The reviewers were all enthusiastic about this paper, noting that the results are well supported by the experimental data. They identified certain areas where they felt that there are limitations in the study and made constructive suggestions for how these could be addressed in a timely fashion.1) The dynamic clamp part is very elegant, but a limitation is that the relation between what happens at the level of the dendritic tip and what is recorded in the soma is complex, especially in starburst amacrine cells (SAC), due to the electrotonic isolation and the differences in selectivity of the different dendrites. When recording from the soma, one looks at an average contribution from the different dendrites, not specifically the ones that connect to the direction-selective ganglion cells of interest. The modelling work of the authors aims at solving this problem, but it is not clear if it really shows that, in such a noisy condition, the voltage in the dendrite will just be the one in the soma up to a scaling factor. Indeed, a major concern among the reviewers is that there is no direct experimental support for the model's prediction that the dendrites attenuate the somatic voltage-clamp signal by a factor of four (Figure 4). This prediction may be essential to interpreting the key experiment. One approach might be a calcium imaging control for voltage attenuation to compare calcium signals in the soma and dendrites in response to somatic depolarization, but that has its own caveats. Another would be simply to apply a range of scaling factors to the command waveform to show that the differences in depression are robust.

Thank you for the helpful suggestions. We agree with the reviewers that the accurate voltage at the dendritic tip is difficult to measure experimentally and we used modeling to estimate the dendritic voltage from our best effort. We followed the reviewers’ suggestion to apply additional scaling factors of 3 and 5 to the command waveforms. We observed similar effects: synaptic depression induced by knockout waveforms was robust with these scaling factors. We have included these results in the revised Figure 5 and the Results section.

In addition, we also compared DSGC IPSCs obtained from voltage-clamping SACs using 4x scaled waveforms with the IPSCs evoked by the actual noisy bar visual stimulation. We quantified the IPSC amplitude and charge transfer, and found that these values in paired recordings are in the lower end of the range for DSGC IPSCs evoked by the noisy bar stimulus (charge transfer: light-evoked: preferred direction: 95.1+/-8.6 pA*s; null direction, 198.6 +/- 22.5 pA*s, 16 cells; paired recordings: 100+ 12.1 pA*s, 22 cells; amplitude: light-evoked: preferred direction: 284.2 +/- 23.2 pA; null direction, 570.3 +/- 52.7 pA; paired recordings: 174.5 +/- 17.9 pA). Considering that more than one SAC form inhibitory synapses onto the DSGC, these values lend support that the measured DSGC inhibitory currents in the paired recordings are within the physiological range. We include these data in the revised Results section.

2) Figure 3 shows that recovery from synaptic depression takes several hundred milliseconds (at 400 ms, most of the paired-pulse suppression is still there), but faster motion stimuli allow only for 100 to 300 ms hyperpolarization (Figure 5D). Thus, one may expect that the suggested mechanism becomes less effective for higher stimulus speeds and that noise-induced suppression of the SAC output should by observed even with intact SAC-SAC inhibition. A more detailed consideration of the time that it takes for synaptic depression to recover is needed to help support the idea that this is indeed at the root of the observed phenomenon and may help understand the limits of the proposed circuit motif. (As a note aside, Figure 5F shows similar levels of noise-induced suppression for different speeds, but it's not clear to what degree the selected SAC voltage traces are representative enough of the ensemble of noise-induced voltage traces to allow for a quantitative comparison.)

We thank the reviewer for raising this excellent point. We have included additional experimental results and discussion to provide a more detailed consideration of the relationship of the surround suppression time window and synaptic depression. We interpret our experimental results below and in the revised manuscript.

In this study, we found that the SAC-SAC-DSGC disinhibitory motif prevents noise-induced synaptic depression. The SAC surround suppression provides a “protective” or “quiet” window to prevent noise-evoked SAC activation and consequent depression from occurring immediately before motion-evoked activation and neurotransmitter release. Noise-triggered SAC activations that occur within the surround suppression time window could engage unwanted depression for the subsequent motion response, because the SAC-DSGC synapse is prone to depression in the sub-second time scale (estimated from the paired pulse experiment in Figure 3). In *Gabra2* cKO mice, this vulnerable time window is no longer kept “quiet” but can be contaminated by noise-evoked SAC activation, leading to unwanted synaptic depression of the SAC-DSGC synapse that we observed during the noisy bar stimulus. In contrast, in WT mice, the SAC-SACDSGC motif actively suppresses the noise response during the time when the moving bar travels across the SAC RF surround. This function does not require that the SAC-DSGC synapse be chronically depressed by the baseline noise before the onset of motion-evoked surround suppression. Therefore, this function does not critically depend on the recovery from existing depression during the surround suppression window.

The reviewers pointed out that at higher speeds, the surround suppression window could be too short to separate the noise response from the motion response, so that synaptic depression occurs even with intact SAC-SAC inhibition in WT mice. We agree with the reviewers on this point, and reasoned that the more depressed motion-evoked release of *Gabra2* cKO SACs is not only due to less recovery from synaptic depression, but is likely also due to further depression by the extra noise activity within the surround suppression window. When we selected *Gabra2* cKO SAC voltage traces at the higher speed that contained the “unwanted” noise activation within the shorter surround suppression window as command waveforms in paired recordings (Figure 5G, lower panel), we observed more severe synaptic depression of the SAC-DSGC synapse than with the control waveform. Therefore, the SAC-SAC inhibition is still effective in shielding the SAC from further depression by the noise activity within the ~150 ms window prior to the motion response. The persistence of the protective role of SAC-SAC inhibition at the higher speed suggests that the short period immediately before the motion response plays an important role in shaping the motion-evoked SAC transmitter release.

The reviewers also raised the question on the selected voltage traces versus the ensemble of voltage traces. We have now included additional analysis of noise-driven activation during the surround suppression window at different speeds in WT and cKO groups, and show that noise suppression is less effective in cKO mice at all speeds (new Figure 5—figure supplement 1). For each speed, we selected SAC command waveforms that matched the average number of noise events and the mean event amplitude of each condition.

We have now added the above results and discussion in the revised manuscript (Results section; Discussion section). Together, these results establish the role of the SAC-SAC-DSGC motif in preventing unwanted noise-induced activity and synaptic depression of SACs during the surround suppression time window.